# Comparative Physiology and Transcriptome Analysis Provides Insights into the Regulatory Mechanism of Albinotic *Bambusa oldhamii*

**DOI:** 10.3390/plants12244090

**Published:** 2023-12-06

**Authors:** Qixia Qian, Quanfeng Ye, Yin Xu, Naresh Vasupalli, Haiwen Lu, Qiutao Hu, Dan Hou

**Affiliations:** 1College of Landscape Architecture, Zhejiang A&F University, Lin’An 311300, China; qian023456@163.com; 2State Key Laboratory of Subtropical Silviculture, Zhejiang A&F University, Lin’An 311300, China; yqf1203299118@163.com (Q.Y.); xuyin-zj@foxmail.com (Y.X.); nareshvasupalli@zafu.edu.cn (N.V.); 2021202011016@stu.zafu.edu.cn (H.L.); 18806507715@163.com (Q.H.)

**Keywords:** *Bambusa oldhamii*, albino, transcriptome, chloroplast and chlorophyll

## Abstract

Albinism is a unique problem encountered in tissue culture experiments, but the underlying mechanism remains unclear in most bamboo species. In this study, we identified the putative regulatory genes in an albino mutant of *Bambusa oldhamii* using comparative physiology and transcriptome analysis. The degeneration of chloroplasts, low chlorophyll (Chl) content and reduced photosynthetic capacity were observed in albinotic *B. oldhamii* compared to normal lines. A total of 6191 unigenes were identified that were clearly differentially expressed between albino and normal lines by transcriptome sequencing. Most genes related to chloroplast development (such as *Psa*, *Psb*) and pigment biosynthesis (such as *LHC*, *GUN4*, *ZEP*) were downregulated significantly in albinotic lines, which might be responsible for the albino phenotype. Moreover, some transcription factors (TFs) such as PIF and GLK1 were identified to be involved in chloroplast development and Chl synthesis, indicating the involvement of putative regulatory pathways *PIF-LHC* and *GLK1-LHC/Psa/Psb* in albinotic *B. oldhamii*. Finally, the downregulation of some stress responsive TFs (like ICE1 and EREB1) suggested a reduction in stress resistance of albinotic *B. oldhamii*. The above findings provided new insights into the molecular mechanism of albinism in bamboo.

## 1. Introduction

The albino mutation is a universal biological phenomenon. According to previous studies, most albino plants have poorly developed chloroplasts [1,2,3]. Chloroplast-encoded genes are transcribed mainly by either multi-subunit plastid-encoded polymerase (PEP) or nuclear-encoded polymerase (NEP) or their cooperation [4]. The PEP is a bacterial-type multisubunit enzyme composed of core subunits (coded for by the plastid *rpoA*, *rpoB*, *rpoC1* and *rpoC2* genes) and additional protein factors (sigma factors and polymerase associated protein, PAPs) [5]. The genetic inactivation of any of the *PAP* genes resulted in an albino phenotype regardless of the potential protein functions [2,6]. According to previous studies, albino leaves usually lack chlorophyll (Chl) and carotenoids (Cars). For example, in rice, iron deficiency significantly depressed expressions of *OsLhca1/2/3/4* (*LIGHT HARVESTING COMPLEX*) and caused a substantial decrease in chlorophyll content and photosynthetic efficiency [7]. The mutation in the carotenoid biosynthesis gene *PDS3* also showed both albino and dwarf phenotypes [8]. In addition, some transcription factors (TFs) were known to regulate chlorophyll biosynthesis and chloroplast development [9]. For example, in *Betula platyphylla*, leaves had decreased chlorophyll content and defective chloroplast development in *BpGLK1*-repression lines [10]. Further, some environmental factors such as light or temperature also induced albinism in plants [11,12], and some albino plants were also more sensitive to different stresses. For example, a mutation in *HSA1* resulted in severe albino phenotype and reduced heat tolerance in rice (*Oryza sativa*) [13].

China has rich bamboo resources, with 43 genera and 500 species, accounting for more than 1/2 of the bamboo species in the world [14]. According to recent reports, the bamboo industry’s production output increased from 82 billion yuan in 2010 to 415.3 billion yuan in 2022, with an average annual growth rate of over 30%. Besides the conventional uses of bamboo in human and animal food, timber, fuel and paper production, it can also be used as an ornamental plant due to its peculiar features, such as different shapes and colors of leaves or internodes [15]. For example, species such as *Phyllostachys nigra* cv. Munro, *Bambusa ventricosa* cv. McClure and *Phyllostachys violascens* cv. Viridisulcata all have high ornamental values because of their black stems, internode structures or internode color variations [14]. Zheng et al. (2021) reported that the abnormal leaves or culms could attract more attentions and enhance the ornamental and economic values of bamboos [16]. Therefore, studying the color variation of bamboo is of great significance.

Albinism is more likely to occur during tissue culture of bamboo. For example, in the tissue culture, green-white striped and albino leaf mutants of *Pseudosasa japonica* were observed [1], and after ^60^Co γ radiation, albino and full-green mutants of *Pleioblastus fortunei* were both derived [17]. Two albino mutants, *ab1* and *ab2,* were identified from *B. edulis* during multiple shoot proliferation [18]. It could be seen that the abnormal chloroplasts had undergone a large-scale deletion and a subsequent loss of chloroplast genes encoding ATP synthases, photosystem II subunits, NADH dehydrogenase, and ribosomal proteins [18,19]. Moreover, a bamboo-specific albino-repressed gene *09A06* was found to be localized to the chloroplasts of *B.edulis* [20]. However, in albino *P. fortune*, no significant difference in gene copy number was found compared with green plants. Only some genes have lower transcription levels, while the transcription level of *psbA* is higher than the wild-type plant [17]. Although a series of studies have been conducted on bamboo albinism, most still concentrate on the study of gene expression related to chloroplast development and Chl metabolism. To date, there are few studies on the putative regulatory network within albino bamboo, which limits the revelation of the bamboo albinism mechanism.

*B. oldhamii* Munro is a member of the subfamily Bambusoideae and is widely planted in in the south of China, like the Fujian, Guangdong and Taiwan provinces. It is an excellent ornamental bamboo species and also a vital vegetable bamboo because of its tender shoots [14]. In the current study, an albino mutant (AL) of *B. oldhamii* was identified during tissue culture, and the regulatory genes related to the albino phenotype were identified through comparative physiology and transcriptome analyses. Our study aims to more comprehensively reveal a common mechanism of albinism in bamboo species. These results will provide new insights into the molecular mechanism of albinism in bamboo.

## 2. Results

### 2.1. Morphological Characteristic and Physiological Variation of Albinotic B. oldhamii

In the current study, we observed that the normal and albinotic *B. oldhamii* pheno-typic characteristics differed entirely (Figure 1a,b): the colors of all tissues on NL plants were light-green to green, but for the AL plants, they were totally white. Moreover, the albino plant showed thinner and curly leaves, and its height and fresh weight was significantly shorter and lower than the normal plant, respectively (Figure 1c,d). The contents of the pigments were then measured. As a result, the content of Chl a, Chl b and Cars were all significantly decreased in albino leaves (0.051 ± 0.017, 0.080 ± 0.025 and 0.020 ± 0.0029 mg/L), which were far lower than that in normal leaves (3.83 ± 0.141, 1.40 ± 0.090 and 0.93 ± 0.029 mg/L) (Figure 1e). Furthermore, the Chl fluorescence characteristics of normal and albino plants were also determined. As shown in Table 1, the values of φPo (*Fv/Fm*) and φEo in AL leaves was about 0.11 and 0.048, which decreased by approximately 85% and 74% compared with that in NL leaves (0.69 and 0.18), respectively. Moreover, the values of ABS/RC, DIo/RC, TRo/RC, ETo/RC in AL leaves were significantly higher than that in NL leaves. The above results indicated that the capacity of PS II reaction center was clearly decreased, thereby leading to serious damage to the photosynthesis of albino plants.

### 2.2. Abnormal Chloroplast Development in Albinotic Leaves of B. oldhamii

To better understand the albinism mechanism in *B. oldhamii*, leaf development was observed in normal and albino strains. There were no serious differences in the tissue structure and cell type, which were composed of epidermal cells, vascular and mesophyll cells in both NL and AL leaves (Figure 2A,B). However, the morphologies of some cells changed in albino leaves. For example, the mesophyll cells were large and loose in the normal leaves, while in albino leaves, they seemed to be small and compact. Moreover, the fusoid cells were shuttle shapes in normal leaves, but they were compressed into flat strip types in the AL leaves (Figure 2A,B). The chloroplast structure in NL and AL leaves was also compared using TEM (Figure 2C–F). The chloroplasts in the normal cells were evenly distributed, with standard shapes, sizes and numbers. The thylakoid and grana showed perfect development in the chloroplasts, and the lamellar structure was clear (Figure 2C,D). However, in the AL leaves, the chloroplasts were poorly developed, and most were small. Further, the chloroplasts were disorder and had no obvious thylakoid and grana lamellar structure inside, and the number of osmiophilic bodies increased (Figure 2E,F). Therefore, the abnormal development of chloroplasts should be the main reason for albinism in ALs.

### 2.3. Overview of Transcriptome Data

To reveal the molecular events occurring in the albinotic bamboo, transcriptome sequencing was performed in NL and AL leaves of B. oldhamii. A total of 115,780,604 and 117,213,598 high-quality clean reads were obtained from three biological replicates of NL and AL leaves, respectively. The average GC content was 49.94% and 48.41%, and the average Q20 was about 98% in NL and AL leaves (Appendix A). The above results indicated that the sequencing quality can meet the requirements of subsequent research.

According to DEG analysis, a total of 6191 unigenes were identified between AL and NL leaves. Among these, 2712 unigenes were upregulated and 3479 unigenes were downregulated significantly in AL leaves (Appendix A). The results of the GO analysis indicated that most DEGs were related to the chloroplast thylakoid membrane or the chloroplast envelope, an integral component of plasma membrane, suggesting their important roles in chloroplast development (Appendix A). Moreover, according to KEGG analysis, most DEGs were involved in the pathways of photosynthesis–antenna proteins, starch and sucrose metabolism, photosynthesis, phenylpropanoid biosynthesis and oxidative phosphorylation (Appendix A).

### 2.4. DEGs Analysis

#### 2.4.1. The DEGs Involved in the Chloroplast Development and Chlorophyll Metabolism

The expression of the DEGs involved in chloroplast development was determined (Figure 3a). As a result, five phytochrome-interacting factors (*PIFs*) were identified that were differentially expressed between ALs and NLs, including *PIF2*, *PIF3, PIF4* and two copies of *PIF1*(*PIF1-1*, *PIF1-2*). Compared with NLs, *PIF1-1* and *PIF1-2* were both downregulated, while *PIF2*, *PIF3* and *PIF4* were significantly induced in ALs. Moreover, two *GLK* genes were observed as DEGs. In ALs, *GLK1* was downregulated while *GLK2* was upregulated compared with NLs, suggesting their putative different roles in regulating chloroplast development and Chl biosynthesis. Finally, the expression patterns of other chloroplast genes were also determined. Among these, the genes that depend on the PEP and NEP were all downregulated significantly in ALs. Taken together, the above key DEGs might be responsible for the abnormal chloroplast development in ALs.

As the content of Chl was clearly lower in ALs than in NLs, the genes involved in Chl metabolism were also focused on. Compared with NL, the genes involved in Chl synthesis (like *UROS*, *CHLM*, *DVR* and *CAO*) were clearly downregulated in ALs, while those involved in Chl degradation (including *HYC1* and *HCAR*) were significantly upregulated (Figure 3b). These results indicated the Chl biosynthesis was inhibited and Chl degradation was promoted in ALs, probably leading to a low Chl content in albino leaves.

#### 2.4.2. The DEGs Involved in Car Biosynthesis

To better understand the regulatory mechanism of albinism in ALs, the Car metabolism pathway was also studied (Figure 4). In the Car pathway, eight DEGs were determined between ALs and NLs, including three *PSYs*, two *β-Ohases*, one *LCY*, *ZEP* and *NCED*. Among these, *PSY*, *β-Ohase*, *LCY* and *ZEP* were key genes encoding the enzymes involved in Car biosynthesis. All of them were clearly downregulated in ALs, indicating that Car biosynthesis was repressed. The *NCED* plays an essential role in regulating Car degeneration. Interestingly, it was also downregulated significantly in ALs. Therefore, the above result suggests that the whole Car pathway seems to be inhibited in albino bamboo, leading to a low content of Cars in ALs.

#### 2.4.3. The Differentially Expressed TFs between ALs and NLs

The differentially expressed transcription factors (DEFs) were also analyzed to explore the regulations of biological processes (Figure 5). Here, approximately 62 TFs were identified as DEFs between ALs and NLs, and the majority of them belonged to the bHLH, bZIP, WRKY and MYB gene families (Figure 5a). The expressions of 30 TFs, such as ICE1, EREB1, et al., were decreased significantly in ALs compared with NLs, (Figure 5b). Most of them were involved in regulating stress response, indicating decreasing stress resistance in albino bamboo.

### 2.5. Validation of the Expression of Key Genes by qPCR

To confirm the reliability of the RNA-Seq data, the expressions of some selected genes were verified with qRT-PCR. As shown in Appendix A, the expression trends of all selected genes were similar to those of RNA-seq, indicating a high rate of consistency between RNA-Seq and qRT-PCR.

## 3. Discussion

The albino mutation is a universal biological phenomenon in the plant kingdom. As in previous studies, these albinism phenotypes were produced by similar variations in chloroplast development, pigments biosynthesis and photosynthesis [2,7,8,9,10]. In bamboos, some candidate genes have been reported to be related to albinism phenotypes in leaves, but most of them were involved in chloroplast development and Chl metabolism [1,18,19,20]. In this study, we mainly focused on albinism in *B. oldhamii*, and explored genes not only related to Chl but also involved in Car biosynthesis and stress responses using the transcriptome method. This method can provide new insights aiding our understanding of the underlying mechanism behind bamboo albinism.

As in previous studies, the variations in the microstructure of chloroplasts in ALs should be the main reason for the low pigment contents of *B. oldhamii* [1,17,18,19,20]. Here, expressions of *PIF2*-*PIF4* were significantly induced in ALs. *PIF* is a negative regulator of chloroplast development, chlorophyll biosynthesis and leaf senescence [22,23,24]. For instance, PIF3 can form an interdependent module with EIN3 and repress the expression of most *LHC* genes [25]. In this study, *LHC* and *LHCa* were also found to be downregulated significantly in ALs, indicating a conserved regulation relationship between *PIF3* and *LHC* to regulate chlorophyll degradation in albino *B. oldhamii*. Moreover, under shade, *PIF* can repress the expression of *GLK* and its target genes by disrupting the dimerization of *GLK* [26]. *GLK1* is a positive regulator of chlorophyll biosynthesis and chloroplast development. In Birch (*Betula platyphylla* × *B. pendula*), the *BpGLK1*-repression lines had yellow leaves, and the expression of genes related to antenna proteins, chlorophyll biosynthesis and photosystem subunit synthesis (like most *LHC*, *Psa*, *HEMA1*, *GUN4* and *CRD1*) were downregulated [10]. In the current study, *GLK1* and its putative targets *PsaK*, *PsaN*, *PsaG*, *PsaH* and *GUN4* were all clearly downregulated in ALs. This suggests that *GLK1* might function as a key negative regulator, leading to defective chloroplast development and decreased chlorophyll content and photosynthesis by repressing downstream target genes in albino *B. oldhamii*. Furthermore, bamboo underwent a whole-genome duplication, resulting in more gene numbers and even new gene functions [27]. For example, *BtFD1* and *BtFD2* were homologous but showed different expressions in flower and vegetative development, and *BtFD1* overexpression led to dwarfisms and apparent reduction in length, but no visible phenotype was observed for *BtFD2* overexpression [28]. In this study, we found that *PIF1-1* and *PIF1-2* were downregulated while *GLK2* was upregulated significantly in ALs, which were contrary to their homologous genes. It suggests that the *PIF1-1*, *PIF1-2* and *GLK2* may have different functions in regulating albinism in *B. oldhamii*, which deserve more experimental verifications in the future.

In addition to Chl, the content of Cars was also reduced significantly in ALs. Carotenoids are mostly C40 terpenoids, a class of hydrocarbons that participate in various biological processes in plants, such as photosynthesis, photomorphogenesis, photoprotection and development [21]. In this study, key genes involved in carotenoid biosynthesis were significantly downregulated in ALs, indicating that carotenoid biosynthesis was seriously inhibited in albino *B. oldhamii* (Figure 4). *ZEP* catalyzes the conversion from antheraxanthin to violaxanthin. The loss of the function of *ZEP* in the *aba1* mutant of *Arabidopsis* and *aba2* mutant in tobacco causes the accumulation of high zeaxanthin levels in leaves and lower ABA [29,30]. *NCED* is a key enzyme regulating ABA biosynthesis; the overexpression of *OsNCED3* in wild-type *Arabidopsis* plants resulted in an increased accumulation of ABA [31]. Therefore, the low expressions of *ZEP* and *NCED* indicated decreased ABA content in ALs. As reported previously, the albino *viviparous12* maize mutant was deficient in ABA [32], and exogenous ABA up-regulates the expression of most plastid genes in albino *Triticum aestivum* [33]. Therefore, the downregulations of *ZEP* and *NCED* might reduce the content of ABA, thus causing albinism in *B. oldhamii*.

Finally, the albino mutants were known to be sensitive to environmental stresses. For example, the loss of the function of *TSV2* in rice led to an albino phenotype and early death regardless of growing temperatures [34]. In this study, *EREBP1*, *ICE1* and some stress responsive genes were lowly expressed in AL. *ICE1* is an upstream regulator of cold-responsive genes in most plants [35,36,37]. Ectopic expression of *AtICE1* and *OsICE1* can delay stress-induced senescence and improve tolerance to abiotic stresses in tobacco [38]. It indicates a decreased cold resistance in albinotic lines. Additionally, we found most *bHLHs* were significantly downregulated in ALs, such as *bHLH112*, *bHLH148*. In Arabidopsis, *AtbHLH112* was positively correlated with salt and drought tolerance [39], and transgenic overexpression of *OsbHLH148* in rice can confer plant tolerance to drought stress [40]. Taken together, the downregulations of *EREB1*, *ICE1* and most *bHLH* genes might lead to a decline in biotic and abiotic stress resistances in albino bamboo.

## 4. Conclusions

In summary, the normal and albinotic lines of *B. oldhamii* were compared and analyzed in this study. Defective chloroplast development and decreased pigment contents were all observed in ALs, resulting in decreased photosynthesis and an albino phenotype of *B. oldhamii*. DEG analysis based on transcriptome sequencing indicated that most genes involved in chloroplast development (like *LHC*, *Psa*, *Psb*, *GUN4*) were significantly downregulated in ALs, which may be regulated by PIF, GLK1 and other transcription factors. Moreover, reduced resistance to stress might also occur in albino *B. oldhamii* due to the low expressions of key genes involved in stress response (including *ICE1*, *EREBP1*) (Figure 6). Our finding identified more candidate genes that participate in the albinism process of *B. oldhamii*, which will facilitate an improvement in the future cultivation of other bamboo species.

## 5. Materials and Methods

### 5.1. Plant Materials

The albinol mutants were derived from different normal multiple shoots incubated in MS medium supplemented with 1 mg/L 6-BA and 0.5 mg/L KT. After separation, the multiple shoots of normal and albinotic *B. oldhamii* were incubated in MS medium supplemented with 0.5 mg/L KT for proliferation. The plants were maintained in a tissue culture room under a 16/8 h (light/dark) photoperiod and were subcultured once a month. The leaves of albino and normal lines were selected as samples and immediately frozen in liquid nitrogen, then stored at −80 °C. Three biological replications were used in each assay.

### 5.2. Pigments Determination

The content of chlorophyll and carotenoids was measured according to the protocol reported previously [41]. About 500 mg of normal and albino freeze-dried leaf samples were used for extraction. The extracts were filtered with filter paper, and the absorbance was determined at 470, 645 and 663 nm using a spectrophotometer. Three biological replicates were used for each biochemical assay. Finally, the content of chlorophyll and carotenoids were estimated using the following equations:The concentration of chlorophyll a (Chl a, mg/L) = 9.784 OD663 − 0.990 OD645(1)
The concentration of chlorophyll b (Chl b, mg/L) = 21.426 OD645 − 4.650 OD663(2)
The concentration of carotenoids (Car, mg/L) = 4.695 OD470 − 0.268(Chl a + Chl b)(3)

### 5.3. Photosynthetic Characteristics Determination

The rapid chlorophyll fluorescence parameters and induced kinetic curve were measured using the plant efficiency analyzer M-PEA (Hansatech, King’s Lynn, UK). The OJIP curve was induced with 5000 μmol m^−2^ s^−1^ red light, the measurement time was 1 s. The data were recorded and the OJIP curve was standardized according to the method of Schansker et al. (2006) [42]. Three biological replicates were used for each assay.

### 5.4. Transmission Electron Microscopy (TEM) Observation

For TEM observation, leaf samples were cut into small pieces (1 mm × 2 mm), fixed in 2.5% (*v*/*v*) glutaraldehyde in 0.1 M phosphate buffer (pH 7.4) at 4 °C overnight, then rinsed and incubated for 1–2 h in a solution of 1% (*w*/*v*) OsO_4_ at 4 °C. The mixtures were subsequently dehydrated using an ethanol series (30%, 50%, 70%, 80%, 90%, 95%) for 15 min, and then infiltrated in a gradient series of epoxy resin. After the embedding, thin sections (70–90 nm) were made with a LEICA EM UC7. After double staining with uranyl acetate 50% ethanol saturated solution and lead citrate for 10 min, the samples were observed under a Hitachi H-7650 transmission electron microscope.

### 5.5. RNA Sequencing (RNA-Seq)

Total RNA was isolated from the bamboo leaves using RNAiso plus (Takara), and three biological replicates were used for each sample. After quality detection, the construction of the library and RNA-Seq was performed using Illumina Novaseq™ 6000. After removing the joints, the raw data were filtered using fqtrim to obtain the clean data. At the same time, Q20, Q30, GC-content and sequence duplication level of the clean data were calculated to ensure all the downstream analyses were based on clean data with high quality. Genes with an adjusted log2 fold change ≥ 2 and *p*-value < 0.05 were assigned as differentially expressed. The assembled transcriptome data was mapped with NCBI_NR (http://ftp.ncbi.nlm.nih.gov/blast), Swiss prot (http://ftp.uniprot.org/pub/databases), eggNOG (http://eggnogdb.embl.de/4.5), Pfam (http://pfam.xfam.org/2017.07), Gene Ontology (GO) (http://geneontology.org/2016.04) and KEGG (http://www.kegg.jp/kegg) for the (differentially expressed genes) DEGs function, annotation and pathways analysis, all website links accessed on 23 February 2023.

### 5.6. qPCR

The extracted RNA leaf samples were converted into cDNA using a PrimeScript RT reagent Kit with gDNA Eraser kit (Takara), and then the cDNA was used as template for qPCR, respectively. The qPCR reaction was performed using the SYBR Premix Ex TaqTM (Takara), and the program underwent denaturation at 95 °C for 10 s, annealing at 60 °C for 20 s and extension at 72 °C for 20 s for a total of 40 cycles. The dissolution curve was determined from 60 °C to 95 °C, and each reaction was set to repeat for 3 times. The NTB gene was set as reference gene [43], and the relative expression was calculated using the 2^−ΔΔCT^ method. The primers used in this study are shown in Appendix A.

## Figures and Tables

**Figure 1 plants-12-04090-f001:**
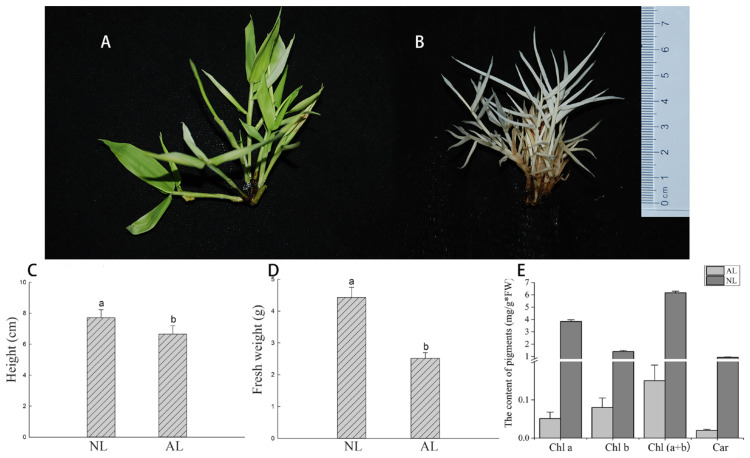
The morphological and physiological characterization of albino and normal lines in *B. oldhamii*. (**A**): normal lines (NLs), (**B**): albino lines (ALs), (**C**): The average heights of Nls and ALs, (**D**): The average fresh weights of NLs and ALs, (**E**): The contents chlorophyll a (Chla), chlorophyll b (Chlb), Chl (a + b) and carotenoid (Car). Different low case letters above columns indicate statistical differences at *p* < 0.01.

**Figure 2 plants-12-04090-f002:**
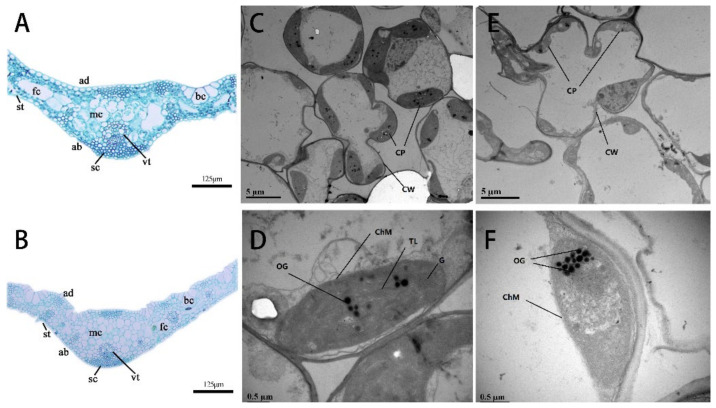
The structures of leaves and chloroplast in normal and albinotic lines in *B. oldhamii*. (**A**): the structure of normal leaves (NL), (**B**): the structure of albino leaves (AL), ab: abaxial epidermis, mc: mesophyll cell, sc: sclerenchyma, bc: bulliform cell, fc: fusoid cell, st: stoma, vt: vascular tissue. (**C**,**D**): the chloroplast microstructure of NL, (**E**,**F**): the chloroplast microstructure of ALs. CW: cell wall, CP chloroplast, TL: thylakoid lamella, OG: osmiophilic granule, ChM: chloroplast membrane, G: grana.

**Figure 3 plants-12-04090-f003:**
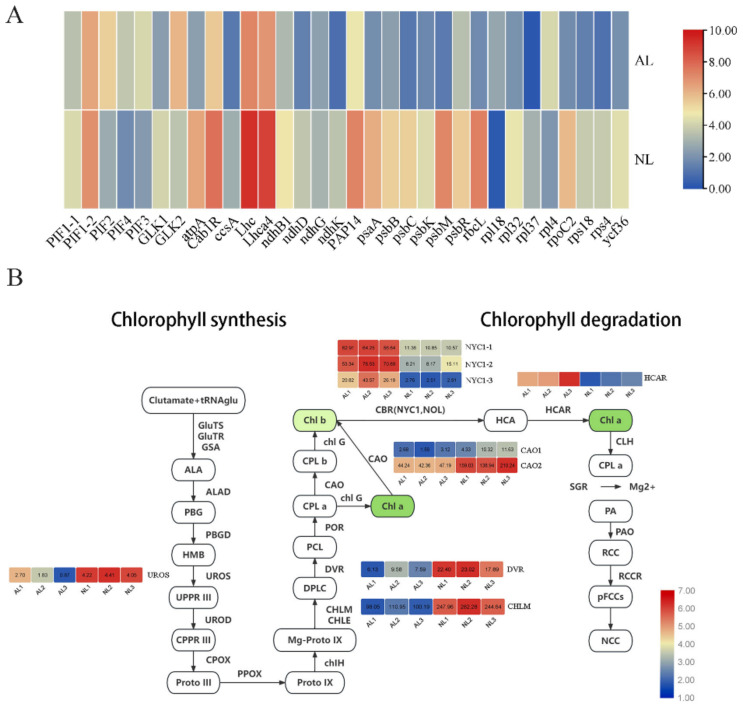
The expression patterns of genes involved in the chloroplast development (**A**) and chlorophyll metabolism (**B**) in normal and albino *B. oldhamii*. Glutamyl-tRNA synthetase (GluTS), Glutamyl-tRNA reductase (GluTR), Glutamate-1-semialdehyde 2,1-aminomutase (GSA), Delta-aminolevulinic acid dehydratase (ALAD), Porphobilinogen deaminase (PBGD), Uroporphyrinogen-III synthase (UROS), Uroporphyrinogen decarboxylase (UROD), Oxygen-dependent coproporphyrinogen-III oxidase (CPOX), Protoporphyrinogen oxidase (PPOX), Magnesium-chelatase subunit ChlH (chlH), Magnesium protoporphyrin IX methyltransferase (CHLM), Magnesium-protoporphyrin IX monomethyl ester [oxidative] cyclase (CHLE), Divinyl chlorophyllide a 8-vinyl-reductase (DVR), Protochlorophyllide reductase (POR), Geranylgeranyl diphosphate reductase (GGDP), Chlorophyll synthase (chlG) and Chlorophyll a oxygenase (CAO). chlorophyll b reductase (CBR: NON-YELLOW COLORING 1 {NYC1) and NYC1-Like (NOL)}, 7-hydroxymethyl chlorophyll a reductase (HCAR), STAY-GREEN (SGR), pheophytin pheophorbide hydrolase (PPH), pheophorbide a oxygenase (PAO) and RCC reductase (RCCR) [4].

**Figure 4 plants-12-04090-f004:**
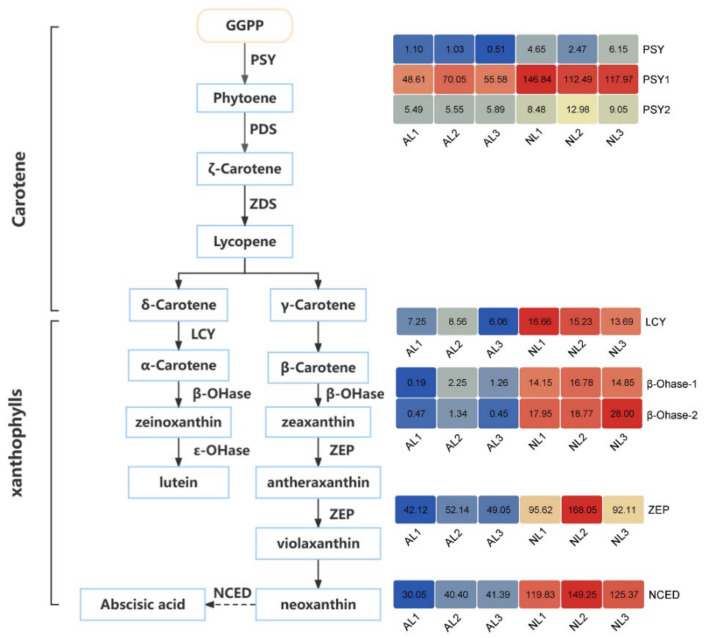
The expression patterns of genes involved in the carotenoid biosynthesis in normal and albino *B. oldhamii*. PDS: phytoene desaturase, PSY: phytoene synthase, ZDS: z-carotene desaturase, LCY: cyclase, β/ε-OHase: β/ε-carotene hydroxylase, ZEP: zeaxanthin epoxidase, NCED: 9-cis-epoxycarotenoid dioxygenase [21].

**Figure 5 plants-12-04090-f005:**
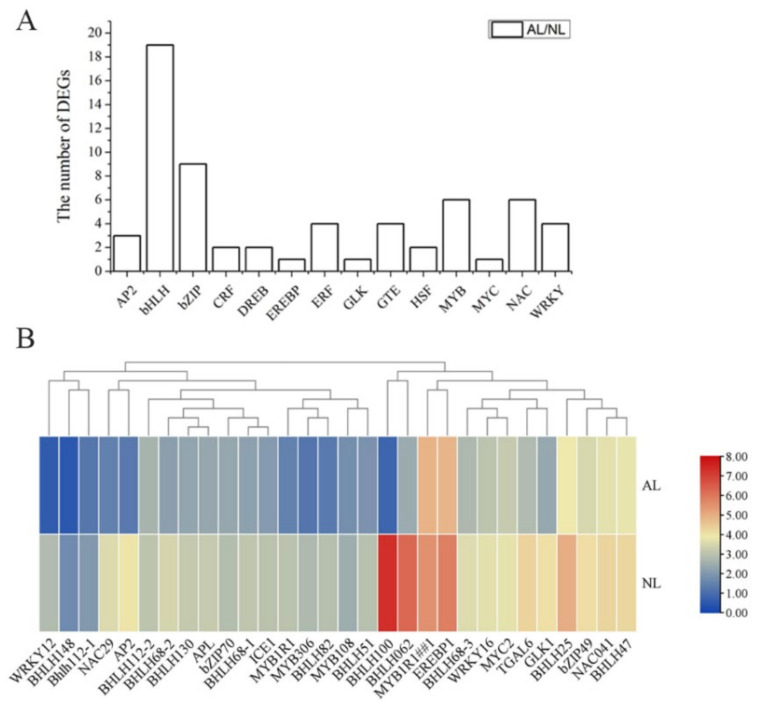
The number (**A**) and expression patterns (**B**) of the differential expressed TFs in normal and albino *B. oldhamii*.

**Figure 6 plants-12-04090-f006:**
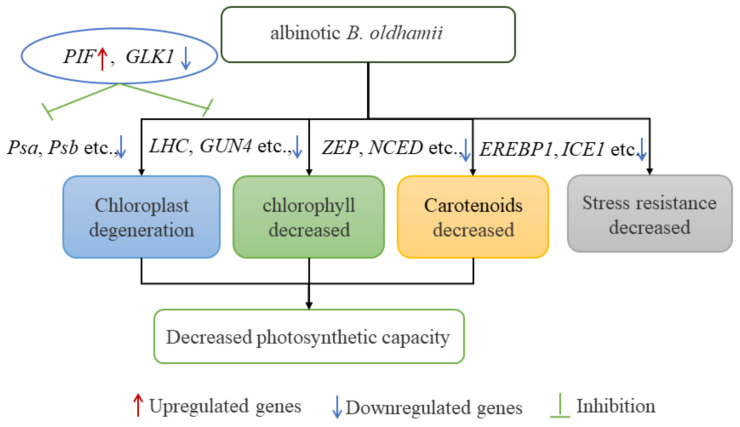
The putative module in regulating albinotic phenotype of *B. oldhamii*.

**Table 1 plants-12-04090-t001:** Chlorophyll fluorescence parameters of normal (NL) and albino (AL) *B. oldhamii*.

	NL	AL
ABS/RC	3.71 ± 0.19 a	417.06 ± 6.88 b
DIo/RC	1.27 ± 0.08 a	404.95 ± 4.48 b
TRo/RC	2.56 ± 0.028 a	6.11 ± 0.36 b
ETo/RC	0.20 ± 0.034 a	12.58 ± 1.03 b
φPo	0.69 ± 0.037 a	0.11 ± 0.014 b
φEo	0.18 ± 0.037 a	0.048 ± 0.047 a

Note: RC: reaction center, ABS/RC Light energy absorbed by RC, DIo/RC: energy dissipated by heat, TRo/RC: energy captured by RC, ETo/RC: energy used for electron transfer, REo/RC: φPo: maximum photochemical efficiency of PS II, φEo: quantum ratios for electron transfer. Different low case letters above columns indicate statistical differences at *p* < 0.01.

## Data Availability

The data presented in this study are openly available in FigShare at https://doi.org/10.6084/m9.figshare.24711213.

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
