# Peer review of "Comparative Physiology and Transcriptome Analysis Provides Insights into the Regulatory Mechanism of Albinotic Bambusa oldhamii"

_plants, 2023, doi:10.3390/plants12244090_

Round 1

Reviewer 1 Report

Comments and Suggestions for Authors

The ms plants-2639632 with the title of Comparative physiology and transcriptome analysis provides insights into the regulatory mechanism of albinotic Bambusa oldhamii investigate an important topic, but the authors should revise their ms to make it suitable for publication in such journal of Pants.

L10-12 please reduce this background to one sentence

L18-19, 21, 23: what are etc.?? You should define them or remove it.

The introduction should be revised and the authors should follow the following order such as background, the problem, and how can be solved, then at the end add the aims.

L29-35 please kindly add suitable citations for such long text.

L46, L48 and L 56 please use the format of the journal when you cite the references within the references, see (Yadavalli et., 2012), (Qin et al., 2007) and (Yang et al., 2015)

L78 please remove etc or define it.

Figure 1C: remove “The” from “The content of pigments …”

In Figures and Tables: Please define all abbreviations under the Tables and Figures.

L204-220 What is this?? Last section was 2.6. Validation of the expression of key genes by qPCR but L204-220 section “2.3. Formatting of Mathematical Components” is something else. Authors should revise these sections.

L237 follow the guidelines of the journal Liu et al., (2007)?

Please make all Latin names as italic.

Discussion section should be revised and the authors should make a link between different paragraphs and avoid jumping.

Please add a conclusion section directly after the discussion section.

Good luck

Comments on the Quality of English Language

minor editing is needed

Author Response

Dear Reviewer

Thank you for your warm work and valuable advices to our manuscript ‘Comparative physiology and transcriptome analysis provides insights into the regulatory mechanism of albinotic Bambusa oldhamii’, ID plants-2639632. There is no doubt that these comments are very helpful for improving our manuscript. Now, all the errors were corrected and marked by red color. We hope our manuscript could meet with approval. The responses to the reviewer’s comments are showed in the attached file.

Special thanks to you for your good suggestions.

Best regards

Dr. Hou.

Reviewer 2 Report

Comments and Suggestions for Authors

Authors should emphasize the significance of this study to raise the interest of the readers. Sufficient background information is not provided in the introduction about the linkage between albinism and low-stress tolerance.  A whole section (line 204-220) is irrelevant to the paper. Enough data is not presented to support the discussion. (please find the attached file for highlighted lines)

Comments on the Quality of English Language

There are several grammatically incorrect phrases and sentences.

Author Response

(The authors gave the same response as above.)

Reviewer 3 Report

Comments and Suggestions for Authors

Bamboo is a symbol of China with important ornamental and economic value. It has a certain proportion of albinism during natural growth. The manuscript entitled ‘Comparative physiology and transcriptome analysis provides insights into the regulatory mechanism of albinotic Bambusa oldhamii provides the relative research in albinism. This study analyzed albinism phenotype of Bambusa oldhamii, a species of subfamily Bambusoideae, by using comparative physiology and transcriptome method. The putative regulatory genes associated with the albinism phenotype were revealed, providing some new insights into the molecular mechanism of albinism in bamboos. However, some issues should be stressed:

1.     In the Introduction, the background of bamboo research should be introduced in detail: 

  In the first paragraph, it is recommended to provide more data to prove the importance of bamboo.   

 The early research of bamboo albino also needs to be introduced, more literatures related to bamboo albino should be cited.

2.    In Results and Discussion

In 2.3, more details of chloroplast genome sequencing results should be added, not just the conclusions.

  Line 85, it is better to have data to support the result of “the normal plants showed significantly higher height and heavier fresh weight than the albino lines”.

 Line 228, the genes related to leaf senescence are little.

 Lines 250-268, is there any regulatory relationship between PIF and GLK? If there is, the relative discussion can be added.

The age of plant material should be added in 4.1.

3.  There are some writing problems need to be revised:

 The latin names should be italicized in the Abstract. The authors need to carefully review the entire manuscript and correct similar errors.

Line 89 and line 90, the first occurrences of Chl and Car require full names, the similar errors in the manuscript also need to be checked and corrected.

The full names of genes appeared in Fig.3 should be added, similar errors should be corrected.

  Lines 204-220, delete.

 Line 252, “were first to be focused on”.

The captains of Figures should in the middle of the text.

Line 323, mg*L1

  Line 347, mol L-1

  Line 374, OsO4

Comments on the Quality of English Language

It is linguistically readable.

Author Response

(The authors gave the same response as above.)

Reviewer 4 Report

Comments and Suggestions for Authors

In this manuscript authors reported the characterization of bamboo albino plants derived from in vitro shoot proliferation. Authors reported that albino plants show poor developed chloroplast, and no obvious thylakoids and lamellar structures can be observed. The analysis of chloroplast genome does not show any significant differences between green and albino plants so authors conclude that differences between both genotypes are due to differences on gene expression. However, chloroplast biogenesis takes place through a coordinated expression of nuclear and chloroplast genomes, so it cannot be ruled out that mutations may appear on the nuclear genome during bud induction which would led to the formation of albino plants. This is a common phenomenon that was observed quite frequently in other species and it is a possibility that authors must consider.

The transcriptomic results point to an impairment of chloroplast biogenesis due to the down-regulation of the gene GLK1 in albino plants. Some plants have 2 to 4 redundant GLK genes, so authors should discuss whether B. oldhamii could have several copies of this gene. A similar discussion about PIF genes is advisable since this is a big family of transcription factors (some plant species have up to 15 members of the PIF family).

It is not clear to me what authors consider biological replications of plant material. Since, as authors state, albino plants appear spontaneously when normal shoots are cultivated in MS medium supplemented with phytohormones, I wonder if authors took samples from one of this albino plants or they used three independent albino plants. In the former case, this cannot be considered independent biological replicates, whereas in this late case, authors can not ensure that the mechanisms of albinism are the same on the three plants.

A thorough revision of the manuscript is necessary. Section of 2.3 of the results must be removed and the quality of the figures must be improved. For instance, information in Figure 3 is not readable. Also, the figures legends should be more informative. For instance, in the figure 2 there is not information about which panels correspond to control plants and which to albino plants.

Comments on the Quality of English Language

I identified some misspellings in the manuscript, that can be easily fixed with a careful reading

Author Response

Dear Reviewer

Thank you for your warm work and valuable advices to our manuscript ‘Comparative physiology and transcriptome analysis provides insights into the regulatory mechanism of albinotic Bambusa oldhamii’, ID plants-2639632. There is no doubt that these comments are very helpful for improving our manuscript. Now, all the errors were corrected and marked by red color. We hope our manuscript could meet with approval. The responses to the reviewer’s comments are showed as followed.

Special thanks to you for your good suggestions.

Best regards

Dr. Hou.

Round 2

Reviewer 1 Report

Comments and Suggestions for Authors

The authors somehow revised their ms but they did not improve it significantly. Some comments, particularly the discussion section still need some improvements to make it deeper and avoid repeating your results.

Comments on the Quality of English Language

Minor editing of English language required

Author Response

Dear Reviewer

Thank you for your warm work and valuable advices to our manuscript ‘Comparative physiology and transcriptome analysis provides insights into the regulatory mechanism of albinotic Bambusa oldhamii’, ID plants-2639632. There is no doubt that these comments are very helpful for improving our manuscript. Now, all the errors were corrected and marked by red color. We hope our manuscript could meet with approval. The responses to the reviewer’s comments are showed as followed:

Q: The authors somehow revised their ms but they did not improve it significantly. Some comments, particularly the discussion section still need some improvements to make it deeper and avoid repeating your results.

Minor editing of English language required

Response: Thank you for your comment, we have improved our manuscript by removing some redundancies and adding some deep discussions, moreover, the English writing errors were revised again as you suggested.

Special thanks to you for your good suggestions.

Best regards

Dr. Hou.

Reviewer 2 Report

Comments and Suggestions for Authors

The article has been improved.

Author Response

Dear Reviewer

Thank you for your warm work and valuable advices to our manuscript ‘Comparative physiology and transcriptome analysis provides insights into the regulatory mechanism of albinotic Bambusa oldhamii’, ID plants-2639632. There is no doubt that these comments are very helpful for improving our manuscript. Now, all the errors were corrected and marked by red color. We hope our manuscript could meet with approval. 

Special thanks to you for your good suggestions.

Best regards

Dr. Hou.

Reviewer 4 Report

Comments and Suggestions for Authors

I thank authors for manuscript improvement. There are still few things that should be amended. The most important is that it is very difficult to interpret the Figure 1F.  To my understanding, under normal conditions, plants do not accumulate significant levels of Mg-Proto IX, Proto IX and Pchlide, since this can lead to deadly photooxidative stress. However, according to the results presented in this manuscript green bamboo plants accumulate much higher amounts of chlorophyll precursors. It would be useful that the figure shows the quantification of each compound in the control and albino plants instead of the ratio for a better interpretation. Besides, I do not think that a photometric measurement is the best way to analyze chlorophyll precursors and wonder whether the absorbance could be affected by the influence of other molecules in the solution (i.e. chlorophyll).

Comments on the Quality of English Language

I still think that a revision of the manuscript is necessary. There are spelling mistakes that must be corrected. For instance:

Line 43. “Induce albinism in plants” instead of “Induce albino in plants”

Line 44 & 273. “stresses” instead of “stress”

Line 238. Sentence starting by “Hence, the DEGs …” should be rewritten.

Author Response

Dear Reviewer

Thank you for your warm work and valuable advices to our manuscript ‘Comparative physiology and transcriptome analysis provides insights into the regulatory mechanism of albinotic Bambusa oldhamii’, ID plants-2639632. There is no doubt that these comments are very helpful for improving our manuscript. Now, all the errors were corrected and marked by red color. We hope our manuscript could meet with approval. The responses to the reviewer’s comments are showed as followed:

Q1: It would be useful that the figure shows the quantification of each compound in the control and albino plants instead of the ratio for a better interpretation. Besides, I do not think that a photometric measurement is the best way to analyze chlorophyll precursors and wonder whether the absorbance could be affected by the influence of other molecules in the solution (i.e. chlorophyll).

Response: Thank you for your comment. We also agree that the absorbance might be affected by the other molecules, so we delete the related contents of Figure 1F to avoid this misunderstanding.    

Q2: I still think that a revision of the manuscript is necessary. There are spelling mistakes that must be corrected. For instance:

Line 43. “Induce albinism in plants” instead of “Induce albino in plants”

Line 44 & 273. “stresses” instead of “stress”

Line 238. Sentence starting by “Hence, the DEGs …” should be rewritten.

Response: We checked the manuscript again, the spelling mistakes have been corrected, and some sentences (such as line 238) were deleted already.

Special thanks to you for your good suggestions.

Best regards

Dr. Hou.